# Regulatory Mechanisms That Guide the Fetal to Postnatal Transition of Cardiomyocytes

**DOI:** 10.3390/cells12182324

**Published:** 2023-09-21

**Authors:** Patrick G. Burgon, Jonathan J. Weldrick, Omar Mohamed Sayed Ahmed Talab, Muhammad Nadeer, Michail Nomikos, Lynn A. Megeney

**Affiliations:** 1Department of Chemistry and Earth Sciences, College of Arts and Sciences, Qatar University, Doha P.O. Box 2713, Qatar; 2Department of Medicine, Department of Cellular and Molecular Medicine, Faculty of Medicine, University of Ottawa, Ottawa, ON K1H 8M5, Canada; jweld090@uottawa.ca (J.J.W.); lmegeney@ohri.ca (L.A.M.); 3College of Medicine, QU Health, Qatar University, Doha P.O. Box 2713, Qatar; ot1608746@student.qu.edu.qa (O.M.S.A.T.);; 4Sprott Centre for Stem Cell Research, Ottawa Hospital Research Institute, Ottawa, ON K1H 8L6, Canada

**Keywords:** fetal to postnatal development, signaling pathways (mTOR, Hippo, YAP/TAZ), micro-RNA, therapeutic potential, hypertrophy and proliferation

## Abstract

Heart disease remains a global leading cause of death and disability, necessitating a comprehensive understanding of the heart’s development, repair, and dysfunction. This review surveys recent discoveries that explore the developmental transition of proliferative fetal cardiomyocytes into hypertrophic postnatal cardiomyocytes, a process yet to be well-defined. This transition is key to the heart’s growth and has promising therapeutic potential, particularly for congenital or acquired heart damage, such as myocardial infarctions. Although significant progress has been made, much work is needed to unravel the complex interplay of signaling pathways that regulate cardiomyocyte proliferation and hypertrophy. This review provides a detailed perspective for future research directions aimed at the potential therapeutic harnessing of the perinatal heart transitions.

## 1. Introduction

Heart disease remains the leading cause of death and disability worldwide, a fact that necessitates ongoing efforts to fully understand the mechanisms that manage the heart’s growth, repair, and eventual dysfunction [1]. Past and prevailing theories on mammalian myocardial regeneration have relied primarily on the promotion of proliferation and reprogramming of adult cardiomyocytes, along with the participation of resident and recruited stem cells. However, it is important to acknowledge that these processes are frequently marked by inefficiency and have had a limited effect on the restoration of myocardial function following cardiomyocyte death [2,3,4,5]. The complex intricacies involved in the morphogenetic process of cardiac histogenesis are a likely contributor to the obstacle associated with encouraging myocardial regeneration in a damaged post-mitotic heart. This is most evident from the failure of employing exogenous stem cells to repair damaged heart tissue. As a result, the occurrence of “restitutio ad integrum” in necrotic cardiac areas is infrequently reported by pathologists, as the predominant outcome is the development of normal scar tissue [6,7,8,9]. A notable disparity is observed in the morphogenetic processes of damaged hearts in zebrafish and salamanders, which results in complete restoration of the heart’s morphology and function. However, the mammalian perinatal heart retains a capacity for restitutio ad integrum post-injury, which is lost in the post-mitotic cardiomyocyte.

The high-resolution day-by-day investigation that developmental biologists have performed on the embryonic heart has led to the identification of transcription factors that control cell-type specification, migration, and differentiation [10,11]. The proliferation of heart cells is the primary mechanism through which the heart expands during the development of a mammalian embryo. During the postnatal period, cardiomyocytes lose their ability to re-enter the cell cycle, and future heart size is primarily due to the process of cardiomyocyte hypertrophy. However, the developmental program that helps facilitate the transition of fetal cardiomyocytes into their postnatal state remains poorly understood. The purpose of this review is to provide a summary of recent research that examines the biology of this transitional program and determine whether these discoveries may offer targets for prospective therapeutic interventions.

## 2. Perinatal Heart Development

Embryonic cardiomyocytes have a high proliferative capacity during fetal development and readily multiply to form a functional heart. Cardiomyocytes undergo a final round of karyokinesis in the absence of cytokinesis at birth to become binucleated, and their proliferative capacity is lost. During this period, cardiomyocytes transition from hyperplastic to hypertrophic growth following binucleation. The size of the mouse heart increases dramatically during the first ten days of life. Individual cardiomyocytes dramatically increase in size and show significantly more cellular structure with muscle striation 10 days post-birth compared with 3 days [12]. This cellular and organ-level hypertrophy during early life is physiologic and scaled to ensure cardiac output matches the organism’s postnatal growth. This is reflected in the stable heart-weight-to-body-weight ratio in the postnatal mouse [12]. Linking these two distinct cellular phenotypes, the fetal heart development program versus the postnatal heart program creates a neonatal transition with distinct structural and presumably distinct gene expression programs (Figure 1).

### 2.1. Cardiomyocyte Cytokinesis, Ploidy, and Loss of Proliferative Capacity

Shortly after birth, mouse cardiomyocytes undergo one final round of karyokinesis without subsequent cytokinesis, resulting in two nuclei in 95% of rodent cardiomyocytes (binucleation) [13]. A diploid genome is found in each of these nuclei. The majority of cardiomyocytes in rodents are binucleated by day 7 [14]. While up to 95% of cardiomyocytes in mice are binucleated shortly after birth, the binucleation index varies significantly between species. For example, the binucleation index of cardiomyocytes in humans has been reported to range between 25% and 60%.

On the other hand, human cardiomyocytes contain four copies of the genome, with mononucleate human cardiomyocytes having one tetraploid nucleus rather than two diploid nuclei, and cell cycle re-entry is also inhibited [15]. Interestingly, zebrafish cardiomyocytes have a single, diploid nucleus and can proliferate throughout their lives. Adult zebrafish hearts, interestingly, can regenerate and heal completely after cardiac injury. Researchers demonstrated that cardiomyocytes disassemble and reorganize their sarcomeres before dividing using a modified cre/lox system in which green fluorescent protein (GFP) was expressed via myosin heavy chain (MHC) [16]. Furthermore, they discovered that Gata4 expression identifies a subpopulation of cardiomyocytes that can re-enter the cell cycle. As a result, the re-expression of fetal proliferative genes in zebrafish results in cardiomyocyte re-entry into the cell cycle to replenish cell populations. This ability, however, is lost in higher vertebrates such as birds and mammals. For example, in vitro stimulation of rat cardiomyocytes with neuregulin-1 NRG-1 increased DNA synthesis, yet only allowed 0.6% of previously non-dividing mononucleate cardiomyocytes to complete cytokinesis [17].

Furthermore, p38 mitogen-activated protein kinase (p38) has been shown to inhibit cardiomyocyte cytokinesis. In vitro, p38 inhibition increased the rate of successful cytokinesis nearly fourfold after FGF1 stimulation compared with FGF1 alone [18]. Collectively, these observations suggest a link between ploidy and divisibility based on in vitro rat models and the proliferative capacity of the zebrafish heart. However, the binucleation index varies broadly across more diverse species, and other multinucleated and polyploid cells such as hepatocytes, skeletal myocytes, and osteoclasts appear not to adopt a direct ploidy versus divisibility rule.

The transitional program responsible for cell cycle arrest and the disjunction between karyokinesis and cytokinesis is currently unknown, though some genes have been implicated. Anillin, for example, is present during the G1, S, and G2 phases of mitosis and is localized to the cell cortex, which aids in forming the contractile ring required for telophase [19]. Anillin fails to localize to the contractile ring in cardiomyocytes after birth, resulting in asymmetric constriction, defective mid-body formation, and failed cytokinesis [20]. Protein Regulator of Cytokinesis 1 (PRC1) collaborates with Anillin to promote cytokinesis during replication. PRC1 expression declines rapidly after birth and is not found in adult cardiomyocytes because they do not undergo cytokinesis [21]. PRC1 re-expression and proper Anillin localization would be required if the cardiomyocyte cell cycle was to be re-engaged.

Another recent study [22] reported that RNA-binding protein with multiple splicing (RBPMS) plays a role in heart development. The researchers discovered that RBPMS regulates embryonic cardiomyocyte cytokinesis, and its absence causes cytokinesis failure, premature binucleation, and non-compaction cardiomyopathy. RBPMS is also known to mediate alternative mRNA splicing in the heart, with PDLIM5 being a key splicing target of RBPMS [22]. These findings shed light on the molecular mechanisms that underpin heart development and have implications for better understanding and developing novel treatments for cardiovascular diseases.

The same loss of proliferative capacity occurs in humans during childhood. Researchers were able to determine the average turnover rate of cardiomyocytes in humans using carbon dating, stem cell marker expression, and thymidine-analogue cancer treatments [23]. The first decade following birth shows a steep decline to around 20% turnover, with a renewal rate of around 1% by age 20. By age 70, the turnover rate is less than 0.5% [23]. Similar findings have been found in mouse models, as well as a slight increase in cardiomyocyte regeneration in the border zone after a cardiac injury such as myocardial infarction, with hearts only being able to heal by forming a scar [24]. In contrast, other muscle cell types can increase proliferation in response to injury to regenerate damaged tissue [25].

A study aimed to restore lost heart muscle, an important goal in cardiovascular regenerative medicine [26]. To capture the complex biology of cardiomyocyte proliferation, the researchers developed a high-throughput phenotypic assay platform using rodent whole-heart-derived cells. They used several readouts, including a transgenic H2B-mCherry system, to detect cardiomyocyte nuclei automatically. The researchers discovered species differences and discovered pan-kinase inhibitors 5 and 36 to be potent inducers of endoreplication and acytokinetic mitosis [26]. The researchers also discovered that commonly used p38 MAPK inhibitors may have off-target effects essential for effective cardiomyocyte cytokinesis. Finally, TG003 was identified as a novel candidate for stimulating cardiomyocyte proliferation [26].

What remains more controversial is whether an early post-natal cardiomyocyte population retains mitotic capacity or whether replicating cardiomyocyte progenitors are derived from a resident stem cell population. Porrello et al. (2011) [27] demonstrated that a 1-day-old neonatal murine heart injured by surgical resection could replenish damaged cardiomyocytes and fully heal [28]. Furthermore, a follow-up study that used the same injury and a Rosa26-LacZ reporter locus confirmed that the regenerated myocardium is made up of cardiomyocytes that originated from an MHC-positive lineage [29], strongly suggesting that new cardiomyocytes are formed through cardiomyocyte replication rather than through an intermediate progenitor cell. If the same injury is inflicted on day 7, the cardiomyocytes cannot divide and replenish, and the heart is permanently damaged, with scar formation and decreased functional capacity [27]. This inability to divide after seven days, particularly in adults, becomes a problem when the heart is injured by myocardial infarction or other disease states that result in cardiomyocyte loss. Although some fetal genes are reactivated in disease states [30], cardiomyocytes can still not re-enter the cell cycle to divide and replenish the damaged population.

### 2.2. Growth and Hypertrophy of Postnatal Cardiomyocytes Necessitate Modifications to the Extracellular Matrix

The heart must undergo significant extracellular matrix (ECM) remodeling to compensate for the dramatic increase in cardiac mass during the first ten days of life. Because individual cardiomyocytes grow by up to 30% during the neonatal period, an equivalent increase in supporting matrix must occur for these cardiomyocytes to grow in size.

Cardiac fibroblasts are the primary regulators of ECM remodeling during neonatal heart development through the secretion of various ECM scaffolding and signaling proteins such as collagen, fibronectin, and heparin-binding EGF-like growth factor [31,32]. Cardiac fibroblasts must maintain a delicate balance between providing enough strength for heart contractions and not becoming fibrotic and inflexible [33]. The number of cardiac fibroblasts doubles after birth, and the ECM is actively remodeled to withstand the mechanical stress placed on the ventricles [34]. Cardiomyocytes must express beta-1-integrin to connect to the ECM, as loss of this protein via cardiac-specific deletion leads to reduced cardiomyocyte proliferation and impaired ventricular function [31]. The transcription factors ZEB1 and ZEB2 are involved in the TGF signaling pathway and regulate ECM remodeling during fetal development, as well as the endothelial-to-mesenchymal transition (EMT) phenotype seen in many cancers [35]. ZEB1 and ZEB2 are essential proteins that regulate extracellular matrix degradation and remodeling of the heart.

In a recent study, Wu and colleagues delve into the intricate role of cardiac fibroblasts (cFbs) and the ECM in cardiomyocyte development [36]. The authors propose that postnatal cFbs, through their modulation of the ECM, promote cardiomyocyte binucleation. Interestingly, this promotion of binucleation is attributed to ECM secretion by the cFbs, rather than the secretion of diffusible factors. The study identifies two embryonically enriched ECM proteins—SLIT2 and NPNT (nephronectin)—as promoters of cytokinesis in postnatal cardiomyocytes, both in vitro and in vivo. The study underscores the dynamic nature of the ECM, a complex protein network that evolves as the heart grows, and its crucial interaction with cardiomyocytes. 

The study by Kuwabara et al. underscores and corroborates the pivotal role of fibroblasts and the ECM they produce [37]. The study highlights that fibroblasts and fibroblast-derived ECM components, including fibronectin, collagen, periostin, heparin-binding epidermal growth factor, agrin, and nephronectin, can stimulate cardiomyocyte proliferation and cytokinesis. Furthermore, the ECM secreted by fibroblasts partially governs cardiomyocyte binucleation and cell cycle arrest, indicating a complex interplay between these cellular components during cardiac development. The authors identified additional ECM proteins expressed by perinatal fibroblasts, including tenascin-X, fibrillin-1, and emilin-1, which could potentially enhance cardiomyocyte cell division. Interestingly, the study found that agrin supplementation can augment the proliferation of adult cardiomyocytes and improve heart function post-myocardial infarction. The authors also propose the existence of a specific subtype of cardiac fibroblasts that promotes cardiomyocyte maturation. Lastly, the study notes that while fibroblast loss led to a reduction in many ECM components, these components were not eliminated, suggesting that other cells may also contribute to ECM production. These comprehensive investigations [36,37] underscore the multifaceted role of fibroblasts and the ECM in cardiomyocyte development and maturation.

## 3. Cardiomyocyte Cell Cycle Regulators

Committed cardiomyocyte progenitor cell populations control myocardium expansion before the final morphologic transition to the early postnatal heart. Several pathways that regulate embryonic cardiomyocyte expansion have been identified, and suppressing these same pathways is essential for the transition to hypertrophic growth [38]. Cell cycle progression markers such as the CDK family, MYC, and E2F are downregulated, while p21, p27, and CDK inhibitors are upregulated [39,40,41,42]. However, the number of cardiomyocytes present after the neonatal transition remains stable, with a very limited generation of new cells throughout postnatal and adult life [43]. During this period, polyploid cardiomyocytes lose their proliferative capacity, after which the heart grows primarily through the physiologic hypertrophy of cardiomyocytes and an increase in cardiac fibroblast and endothelial cell populations [44]. Given the central importance of the proliferation to hypertrophy reorientation, a large number of studies have examined the role of cell cycle regulatory factors in this developmental process, and whether it is the primary determinant for cardiomyocyte maturation and terminal differentiation.

### 3.1. Cyclins/CDKs in the Postnatal Heart

Cyclins belong to a protein family that regulates the progression of cells through the cell cycle. Each Cyclin involved in the division is synthesized and degraded during each cell cycle. The cyclins all share a conserved 150 amino acid region known as the “cyclin box”, which binds to the N-terminus of their respective cyclin-dependent kinases (CDKs) [45]. CDKs are cyclin-dependent serine-threonine kinases that become enzymatically active when they interact with cyclins. At specific cell cycle stages, cyclins and CDKs interact to form complexes that drive cells through cell cycle checkpoints [46]. Cyclin and CDK regulation is critical during the hyperproliferative phase of cardiac growth [47,48,49]. 

In vitro, cyclin B1 overexpression in isolated adult rat cardiomyocytes increased total cell number by up to 40%, indicating increased proliferative capacity [50]. Another study using transgenic mouse lines overexpressing cyclins D1 and D3 revealed increased DNA synthesis in cardiomyocytes and decreased infarct size following coronary artery occlusion [51]. Furthermore, cardiomyocyte-specific cyclin A2 overexpression increased cardiomyocyte proliferative capacity after birth, as measured by pH3 staining [52]. Kinase assays and pH/ki67 staining also show an increase in actively cycling cells after isolation and culture. This proliferative ability, however, does not translate in vivo. When overexpressed in mice, cyclin D1 increases DNA synthesis and nucleation but not cardiomyocyte proliferation [53]. 

CDK-Activating Kinases (CAKs) and Cyclin-Dependent Kinase Inhibitors (CDKIs) are also important positive and negative cyclin/CDK activity regulators that control cell cycle progression. After birth, the upregulation of CDKIs coincides with the downregulation of Cyclins and CDKs [47,54,55]. CDKIs are divided into two structurally and functionally distinct families: the INK4 family (p15, p16, p18, and p190) and the Cip/Kip family (p21, p27, and p57). The INK4 family inhibits enzymatic activity by blocking the interaction of Cyclin D and CDK4/CDK6 [56]. The Cip/Kip family of inhibitors selectively inhibits CDK2’s interaction with Cyclin E, thereby inhibiting S-phase progression [57]. Furthermore, they can inhibit Cyclin A and CDK1 activity to play a broader role in mitosis inhibition [57]. P27 and p57 are CKIs that collaborate to promote cardiomyocyte cell cycle exit and terminal differentiation [44,58]. P21 also promotes cell cycle arrest and prevents re-entry [59]. The Cip/Kip family of CDKIs is undetectable during embryonic heart development and begins to increase during the perinatal transition, with adult cardiomyocytes having the highest expression [30]. It has been demonstrated that inhibiting these three CKIs allows rat ventricular cardiomyocytes to re-enter the cell cycle and actively proliferate. This was accompanied by the re-expression of several fetal genes, the downregulation of many adult genes, and a change in cellular morphology [58].

### 3.2. Transcriptional Control of the Cardiomyocyte Cell Cycle

In addition to the general cell cycle regulatory factors, specific transcription factors appear to also exert control over cardiomyocyte cell division. The most studied proteins in this regard are the eight-member E2F transcription factor family. E2Fs retain divergent functions, where individual family members engage in transcriptional activation, while others act as transcriptional inhibitors [56]. Cyclins/CDKs, DNA damage/repair genes, checkpoint genes, and apoptosis genes are their primary genomic targets [60]. Individual E2F family member functions have been difficult to study due to overlapping compensatory roles across the group of eight proteins [61]. Transgenic deletion of E2F3 is lethal to embryos due to congestive heart failure, but targeted deletion of other members has no effect. Conversely, overexpression of E2F1 to 4 increases the rate of S-phase entry in isolated rat cardiomyocytes, while overexpression of E2F1 and three-induced apoptosis [62,63,64].

Three pocket proteins (Rb, p107, and p130) regulate the cell cycle’s G1/S transition by regulating E2F-effectors [65]. Embryonic development is distinguished by increased Rb expression and decreased p130 expression. Rb is important for regulating cardiomyocyte cell cycle withdrawal and differentiation. Rb can bind to E2F to recruit transcriptional repressors and inhibit the G1/S transition when it is dephosphorylated. When phosphorylated by CDK2 and CDK4, Rb is unable to bind E2F and allows the transcription of cell cycle progression genes to occur [66,67]. While Rb deletion causes embryonic lethality, deletion of both Rb and p130 results in increased heart-weight-to-body-weight ratio, cell number, and active proliferation cardiomyocytes, as measured by BrdU incorporation and pH3 staining [68]. 

N-myc, L-myc, and C-myc are transcription factors in the myc family. The myc family interacts with the protein Max to function. Because of their association with increased cellular proliferation, the function of the myc family has been studied primarily in the context of cancer [69,70]. C-myc can mediate G1-phase exit by upregulating Cdk4, Cdc25A, and Cyclins A, D1/2, and E, and antagonizing p27’s actions [69,70]. Transgenic deletion of c-myc causes early embryonic death, but this cannot be attributed solely to heart defects because c-myc is required in the development of many other vital organs and tissues [71]. Transgenic c-myc overexpression causes hyperplastic ventricles during neonatal development, but this increased proliferation subsides during cardiomyocyte maturation [72]. 

Two recent studies showed that activator protein-1 (AP-1) transcription factors are essential in cardiomyocyte development and regeneration [73,74]. AP-1 members not only regulate post-natal cardiomyocyte maturation by activating the expression of fatty acid metabolic genes [73], but they also contribute to extensive changes in chromatin accessibility in regenerating cardiomyocytes after injury, promoting cardiomyocyte dedifferentiation, proliferation, and protrusion into the injured area [74]. These findings highlight AP-1’s potential as a therapeutic target for treating congenital heart diseases and improving cardiac repair and regeneration.

The HIF1 transcription factor has also been shown to play an important role in cell cycle kinetics regulation. HIF1 functions as a c-myc antagonist [75]. In mice, transgenic deletion of HIF1 causes cardiac hyperplasticity, resulting in outflow tract obstruction and functional complications [76]. Interestingly, in adult transgenic mice, HIF1 also induces vascular endothelial growth factor (VEGF), but these transgenic mice do not show increased angiogenesis or cardiomyocyte proliferation [77].

Meis1 is required for p15, p16, and p21 transcription activation and thus plays an essential role in regulating cardiomyocyte proliferation [78]. Although Meis1 gene targeting resulted in death by embryonic day 14.5 due to disrupted hematopoiesis, neonatal cardiac-specific (αMHC) Meis1 deletion increased the proliferative window of cardiomyocytes, as evidenced by increased pH3+, Ki67+, and BrdU+ incorporation. Conversely, Meis overexpression reduced neonatal cardiomyocyte proliferation and inhibited early developmental regeneration of the myocardium [78].

Two recent studies [79,80] focused on the critical period of perinatal heart development and the role of chromatin landscape and transcriptional regulation in shaping the abilities and characteristics of cardiomyocytes. Quaife-Ryan and colleagues’ study presents a comprehensive transcriptomic analysis across a limited time-series analyses (P1, P14, and P56) of multiple cardiac cell types in neonatal and adult mouse hearts, revealing that the neonatal heart has a transient regenerative capacity that is lost due to changes in transcription and chromatin accessibility during postnatal maturation [79]. Zhang and colleagues’ study further illuminates this perinatal transition by identifying key transcription factors, MEF2 and AP1, that drive these phenotypic changes. This study also found thousands of dynamic regulatory elements in perinatal cardiomyocytes that mediated transcriptional reprogramming, facilitating the switch from fetal to neonatal circulation [80]. Together, these studies offer significant insights into the molecular mechanisms governing perinatal heart development, highlighting the role of transcription factors and chromatin architecture in turning on or off the regenerative capacities and functional adaptation of cardiomyocytes.

## 4. Primary Cardiomyocyte Growth Signaling Pathways

### 4.1. The Hippo Pathway

One prominent signal transduction conduit that has been shown to have a dramatic impact on cardiomyocyte growth vs. hypertrophy is the Hippo pathway, originally discovered to regulate organ size in drosophila [81]. This pathway is well-preserved and serves a similar function in animals [82,83]. Mst1 and Mst2 are the human orthologs of Hippo. YAP1 and its paralog TAZ are phosphorylated and sequestered in the nucleus when the Hippo pathway is activated. When the Hippo pathway is shut down, YAP1 is not phosphorylated and remains active in the nucleus, where it can regulate genes like Birc2, Birc5, Cyr6, and Hoxa1 to promote cell proliferation and organ growth [84]. The Hippo pathway can control cell proliferation and organ size by modulating YAP1 expression. After birth, the Hippo pathway is activated to inhibit YAP1 activity and halt cardiomyocyte proliferation [85]. 

Cell growth, proliferation, specification, and differentiation are functions shared by YAP1 and TAZ [86]. Members of the BCL-2 and inhibitor of apoptosis (IAP) families, such as surviving and MCL1, can be upregulated by both. YAP/TAZ is also crucial in organ regeneration and tissue repair, where it influences not only proliferation but also cell survival, dedifferentiation, and stem cell expansion [87]. The Hippo pathway has been previously shown to induce Wnt signaling and increase cardiomyocyte numbers in the developing heart [87,88,89,90,91,92].

YAP1 overexpression during development causes liver and heart overgrowth; it has also been demonstrated to promote cardiomyocyte proliferation and regeneration after injury [93]. In adult mice, YAP1 hyperactivation causes liver overgrowth but not heart overgrowth [94]. Homozygous YAP1 deletion results in death on embryonic day 8.5, whereas mice with TAZ deletions survive [95]. The morula stage is not passed by YAP1 and TAZ double deletions [96]. Defects in angiogenesis, vascularization, and myocardial hypoplasia are caused by cardiac-specific deletion of YAP1 and TAZ [90]. In mice, cardiac-specific deletion of SAV1, a member of the Hippo pathway, reduces YAP1 phosphorylation and, thus increases YAP1 activity [97]. Increased YAP1 transcriptional activation causes embryonic cardiomegaly and neonatal death due to severe heart defects. These findings suggest that the YAP1 protein and the Hippo pathway are involved in a delicate balance of producing a heart large enough to pump blood while matching cardiac growth to whole-body maturation.

### 4.2. The PI3K/PTEN/AKT Pathway

The PI3K/PTEN/AKT pathway regulates cell cycle processes directly by influencing downstream targets such as CREB, p27, and FOXO [98,99]. PI3K converts PIP2 to PIP3, which can then phosphorylate and activate AKT. p-AKT then targets and activates mTORC1, which interacts with p70-S6 kinase to activate transcription and translation [100]. PTEN (Phosphatase and Tensin Homologue on Chromosome 10) is a tumor suppressor with multiple functions. PTEN inhibits PI3K by converting PIP3 to PIP2 via phosphatase activity [101]. This inhibits cell growth by downregulating protein synthesis [102,103]. PTEN conditional deletion revealed additional PTEN functions in the cell-type specification and cardiac muscle contractility [104]. 

In addition, the pathway regulates apoptosis in response to DNA damage [105]. Constitutive activation of protein kinase B (PKB) or Akt increases the molecular half-life of cyclin D [106]. Inhibiting PI3K signaling, on the other hand, increases cyclin D degradation. CDK2 has also been identified as an Akt target during cell cycle progression [107]. The PI3K/AKT pathway reduces P27 expression to promote G2/M progression [108]. Overactivation of the PI3K/AKT pathway in the neonatal heart increases the number of cardiomyocytes [109]. Each regulatory pathway is critical in establishing cardiomyocyte endowment and controlling heart size. These pro-proliferative pathways are suppressed after birth, preventing cardiomyocyte cell-cycle re-entry.

## 5. Epigenetic Regulation of Cardiomyocyte Cell Cycle

Global epigenetic changes occur across the genome after birth, accompanying the transition from a hyperplastic to a hypertrophic heart. According to gene expression profiling (RNA-seq) and genome-wide methylation sequencing, the cardiac methylome changes dynamically during the neonatal period. The regulation of DNA methylation is crucial in suppressing proliferative genes and activating hypertrophic ones. Hypermethylation accounts for up to 80% of methylome changes, resulting in the transcriptional repression of replication and developmental pathways [42]. Furthermore, it was discovered that inhibiting DNA methylation after birth reduced cardiomyocyte binucleation and increased the proliferative index as indicated by phospho-histone 3 (pH3) expression [44]. 

The role of epigenetic regulation via histone modification in neonatal heart transition has also been demonstrated. Histones are hyperacetylated in embryonic cardiomyocytes (H3K9/14, H3K18, H3K27) but become hypoacetylated shortly after birth, coinciding with cell cycle arrest [110,111,112]. Hdac1 and Hdac3 overexpression reduces global acetylation and suppresses cyclin-dependent kinase (CDK) inhibitors, leading to increased proliferation in 1-day-old (1D) hearts [113]. In contrast, cardiomyocyte-specific Hdac3 deletion causes cardiac hypertrophy and metabolic disorders [114]. In mice, global deletion of Hdac1 causes death by postnatal day ten due to defects in cell cycle progression, whereas global deletion of Hdac2 causes unrestricted cardiomyocyte proliferation and death [115,116]. Although cardiomyocyte-specific deletion of Hdac1 or Hdac2 results in no phenotype, a cardiomyocyte-specific double-deletion causes several heart defects associated with cell proliferation [116]. 

Other post-translational modifications include histone methylation, which also serves as an active modifier/repressor of gene expression in cardiomyocytes. Histone methylation (H3K9me3 and H3K27me3) occurs on the promoters of many cell cycle genes after birth [117]. H3K9me3 enrichment on E2F-, Rb-, and SUV39H1-dependent promoters results in long-term silencing of pro-proliferative genes regulated by these factors. Rb and Suv39h1 depletion has also increased cell-cycle gene expression and allowed some cell-cycle re-entry in adult cardiomyocytes [117]. Epigenetic modifications are active at maintaining the adult cardiomyocyte phenotype through suppression of fetal growth and development pathways. This combinatorial epigenetic activity matches the demands of the cardiomyocyte population as it transitions from proliferative to hypertrophic growth strategies [33].

## 6. The Role of Non-Coding RNAs in Heart Development

Recent research suggests that non-coding RNAs (ncRNAs) may be a key driving force in the transition from fetal to adult cardiomyocytes [118,119]. MicroRNAs (miRNAs), PIWI-interacting RNAs (piRNAs), transfer RNAs (tRNAs), ribosomal RNA (rRNA), small nuclear RNAs (snoRNAs), and long non-coding RNAs (lncRNAs) have all been shown to be involved in heart development and regeneration [120]. piRNAs have been shown to aid germline genome stability by silencing repetitive and transposable elements [121]. While the function of specific housekeeping ncRNAs, such as rRNA and tRNA, is well understood, the functions of other ncRNAs are unknown due to their recent discovery. 

LncRNAs are longer than 200 bases and make up most of the non-coding transcriptome. LncRNAs are found in the nuclei and cytoplasm of cells and are subject to post-transcriptional modifications such as splicing, capping, and polyadenylation. The wide variety of functions exerted by lncRNAs is only beginning to be characterized, and several have already been implicated in heart development. Braveheart (Bvht) is a lncRNA that plays a role in differentiating embryonic stem cells into mesodermal progenitors [122]. Bvht acts as an epigenetic regulator, lowering the expression of the Suz12/PRC2 complex. Because the Suz12/PRC2 complex normally inhibits cardiac-lineage-specific gene expression, Bvht activates cardiac-specific genes [122,123]. 

The non-coding developmental regulatory RNA (Fendrr) adjacent to the lncRNA FOXF1 has also been identified as a regulator of cardiac specification. Fendrr is involved in the differentiation of tissues from the lateral mesoderm into the heart. When Fendrr expression is lost, Nkx2-5 and Gata6 expression increases and the embryo dies on embryonic day 13.5 due to heart failure [124]. 

Finally, it was recently discovered that the lncRNA Upperhand (Uph) maintains the super-enhancer signature (H3K27ac) on the upstream regulatory regions of the Hand2 gene. Hand2 expression was inhibited by blocking Uph transcription, resulting in ventricular hypoplasia, outflow defects, and heart failure, leading to embryonic death, similar to the effects of transgenic Hand2 deletion in embryos [125]. The significance of regulatory ncRNAs in heart development is being established. Indeed, we have only begun to understand the function of lncRNAs, while little is known about piRNAs and snoRNAs. 

### 6.1. MicroRNAs in Embryonic Heart Development

MiRNAs are 22-nucleotide-long single-stranded ncRNAs that target messenger RNA (mRNA) at the post-transcriptional level [126,127]. They are currently the most extensively researched ncRNA, primarily acting as negative regulators of gene expression [128]. Primary miRNAs (pri-miRNAs) are transcribed from genomic DNA by RNA polymerase II or produced as by-products from intronic or exonic regions of mRNA transcripts [129]. Drosha and DiGeorge syndrome critical region 8 (DGCR8) cleave Pri-miRNAs into 70 nucleotide precursor miRNAs (pre-miRNAs) [129,130]. The pre-miRNA is exported from the nucleus into the cytoplasm by exportin 5 [131,132]. Another RNAse III enzyme that cleaves the pre-miRNA into the mature miRNA is the protein Dicer. While the other arm of the miRNA is degraded, one arm is loaded onto Argonaute (Ago) proteins to form an RNA-induced silencing complex (RISC). The RISC uses complementary base-pairing to target the 3′ untranslated region (UTR) of mRNA transcripts, preventing translation and promoting degradation [131,133]. Mature miRNAs can also be packaged into vesicles and exported into peripheral tissue and the bloodstream, where they can affect other tissues and cell types [134,135]. 

Micro-RNAs are critical for development. Dicer germline deletion in mice and zebrafish is embryonically lethal, preventing development beyond the gastrulation stage [136,137]. Furthermore, using Cre-mediated deletion under the control of the Nkx2.5 promoter, Dicer deletion during mouse heart development resulted in death due to defective heart morphogenesis [138]. DGCR8 germline deletion caused proliferation defects, dilated cardiomyopathy, and heart failure [139]. Chen et al. strongly support the importance of miRNAs in the neonatal transitional program [140]. Transgenic postnatal cardiac-specific deletion of Dicer using a Cre-Lox system controlled by the MHC promoter resulted in neonatal death on day five due to abnormal expression of cardiac contractile proteins, sarcomeric disarray, slower heart rates, and decreased fractional shortening [140]. 

Several miRNAs have been identified as being involved in heart development. Tightly regulated transcription factor patterning is required for proper cell fate determination, migration, and differentiation of cardiac progenitors. Recent zebrafish research has revealed that miR-138 is required to establish chamber-specific gene expression patterns [141]. Cspg2, Notch1b, and Tbx2 are explicitly expressed in the atrioventricular canal (AVC) of the zebrafish heart, distinguishing it from the atria and ventricles [142,143]. MiR-138 antagomir treatment resulted in the expression of AVC-restricted genes in the atria and ventricles, but ventricular cardiomyocytes did not mature [141]. Future mouse genetic studies will elucidate the role of miR-138 in transcription factor patterning in mammalian heart morphogenesis. 

In the mouse heart, beginning on embryonic day 8.5, miRNA-1 (miR-1) and miR-133 are expressed in cardiac and skeletal muscle and are essential for cardiac development [144,145]. MiR-1 (miR-1-1 and miR-1-2) and miR-133 (miR-133a-1 and miR-133a-2) are produced as pairs from bicistronic transcripts in vertebrate hearts. miR-1-1 and miR-133a-2 are derived from the same intergenic region on mouse chromosome 2, whereas miR-1-2 and miR-133a-1 are derived from an intron of Mib1 on mouse chromosome 18. [146]. MiR-1 and -133 control cardiac specification and growth by targeting Hand2 and Cyclin D and their expression is controlled directly by Mef2, Myocardin, and SRF [35,147]. Surprisingly, miR-133 has been shown to target SRF, resulting in a negative feedback regulation loop [146]. Overexpression of miR-1 under MHC promoter control reduces proliferating ventricular cardiomyocytes, resulting in premature cell cycle exit [35]. Similarly, exogenous miR-1 introduced into Xenopus embryos caused heart development defects [146]. MiR-1 promotes myoblast differentiation in culture, whereas miR-133 promotes proliferation [146]. The same study discovered that miR-1 targets and suppresses HDAC4. HDAC4 inhibits Mef2 transcriptional activation, and thus miR-1 promotes Mef2 target expression [146]. MiR-1 deletion causes heart malformations, cell cycle dysregulation, and electrophysiological defects. Furthermore, around embryonic day 15, half of the null mice die from cardiac malfunction and septal defects, most likely caused by increased Hand2 levels [138]. The other mice that survive gestation die suddenly, most often from arrhythmias. Surviving hearts also have more actively dividing cardiomyocytes and cardiac hyperplasia [138]. 

MiR-133 is a family of two genetically identical members that target Cyclin D2 [147]. Specific deletion of miR-133a-1 or miR-133a-2 results in no phenotype, but a double deletion decreases embryonic survival due to ventricular septal defects [147]. Mice that survive until they are born develop dilated cardiomyopathy and die from heart failure. Overexpression of miR-133, on the other hand, resulted in death on embryonic day 13.5, and embryos showed increased Cyclin D2 levels and decreased cardiomyocyte proliferation as measured by pH3 staining [147]. In vitro overexpression of miR-133 inhibited cardiac hypertrophy, whereas infusion of an miR-133 antagonist into the myocardium caused significant and sustained cardiac hypertrophy [148]. RhoA and Cdc42, which regulate cardiomyocyte growth, are miR-133 hypertrophy targets [148]. Nelf-A/WHSC2, a nuclear factor involved in cardiogenesis, is also targeted by miR-133. 

MiR-143 and miR-145 are found on mouse chromosome 18 and are transcribed as a cluster. Both are expressed in cardiac progenitor cells from embryonic day 7.5 to day 16.5. Following this, they are only found in visceral and vascular SMCs [149,150]. SRF and Myocardin control heart morphogenesis and smooth muscle cell gene expression by targeting CArG boxes found in the upstream enhancer region of miR-143/145 [151]. Elk1, an SRF cofactor, is targeted by miR-143, while Myocardin, Kruppel-like factor 4 (Klf4), and calmodulin kinase II-delta are targeted by miR-145 (Camk2d). While the first two genes (Klf4 and Camk2d) are primarily involved in smooth muscle cell proliferation and differentiation in the heart, they have both been identified as positive regulators of proliferation [151]. Individual or combined deletion of miR-143 or miR-145 causes no obvious phenotype until adulthood. Deficient mice develop hypotension and aortic and femoral artery wall thinning, eventually leading to neointimal lesions in old age. Furthermore, cardiomyocytes exhibit cytoskeletal disarray and reduce migration ability [149,151,152]. 

The primary myosin heavy chain (Myh) gene expressed during embryonic development is the slower-acting MHC (Myh6). Cardiomyocytes use the MHC (Myh6) variant, which has faster contraction kinetics, after birth [153]. Researchers recently discovered the miR-208 family, derived from the intronic regions of these two genes. Myh6 encodes miR-208a, whereas Myh7 encodes miR-208b. This means that the isotype switch of Myh expression in the heart also results in a switch in miRNA-208 expression [154]. Furthermore, it has been demonstrated that the miR-208 family participates in the feedback regulation of their host genes during development and hypertrophy [155,156]. Gata4 and Cx40, which regulate cardiac morphogenesis, are important targets of miR-208a [157]. MiR-208a genetic deletion results in no phenotype, indicating that it is not required for embryonic development or cardiac morphogenesis. These mice, however, were resistant to stress-induced hypertrophy [155,156]. Overexpression of miR-208a, on the other hand, resulted in increased cardiac hypertrophy, conduction defects, and induction of Myh7 expression [155]. The role of miR-208b in cardiomyocytes after birth remains unclear. 

The miR-218 family, which includes miR-218a-1, miR-218a-2, and miR-218b, is highly conserved in vertebrates. The family is located in the intron of slit homologs 2 and 3 (Slit2 and Slit3) and targets Roundabout receptors 1 and 2 (Robo1 and Robo2). miR-218, Slit2, and Robo are required for proper heart tube formation during embryogenesis [158]. Tbx5 is another functional target of miR-218 and their expressions during development are correlated, implying that miR-208 is involved in cardiac specification and differentiation. Tbx5 overexpression and miR-218 downregulation have similar effects, resulting in heart looping defects and ventricle abnormalities [159,160]. Furthermore, miR-218-1 inhibition can reverse the cardiac defects caused by Tbx5 overexpression [159]. 

The miR-17-92 cluster is a polycistronic gene that produces six miRNAs (miR-17, miR-18a, miR-19a, miR-19b-1, miR-20a, and miR-92-1) from a single transcript [161]. Because of their role in regulating proliferation and differentiation, they have received the most attention in the context of cancer research and are also known as Oncomir-1 [162]. According to recent research, the cluster affects cardiac development by targeting the 3′-UTR of Isl1 and Tbx1. The genetic deletion of the miR-17-92 cluster resulted in the failure of Isl1 and the downregulation of Tbx1 during embryonic development [163]. MiR-17-92-cluster-deficient mice die during infancy due to lung hypoplasia, thin ventricle walls, and ventricular septal defects [164]. Myocardial differentiation is inhibited in Bmp-deficient mouse embryos, and the expression of the miR-17-92 cluster is downregulated [163]. MiR-17-92 cluster overexpression inhibits organ growth and hematopoietic cell differentiation. Overexpression of the miR-17-92 cluster also reduces the number of cardiac progenitors in the second heart field, resulting in outflow tract defects by directly targeting Isl1 and Tbx1 [163].

### 6.2. MicroRNAs Affect Cardiomyocyte Proliferation in the Perinatal Heart

Recently, a link between miRNAs and the Hippo pathway has been discovered during heart development. The miRNA-302-367 cluster is made up of eight co-transcribed polycistronic miRNAs (miR-302a, 302a*, 302b, 302b*, 302c, 302c*, 302d, and 367) [165]. The cluster is expressed in embryonic stem cells, where it promotes cell proliferation while maintaining a dedifferentiation phenotype. This cluster has also been shown to regulate cardiomyocyte proliferation during embryonic heart development. Transgenic cardiac-lineage-specific (Nkx2-5 Cre) miR-302-367 cluster deletion resulted in no discernible phenotype. However, transgenic overexpression had a greater impact, with miR-302-367 overexpression in cardiac-lineage-specific cells resulting in significant cardiomegaly due to increased cardiomyocyte proliferation [91]. MiR-302-367 was found to target the 3′ UTR regions of several Hippo pathway regulators, including Mst1, Lats2, and Mob1b. Overexpression of miR-302-367 decreased Hippo pathway expression and increased YAP1 nuclear localization, resulting in the death of the mice by postnatal day 28 [91]. 

The miR-15 gene family has also been linked to neonatal cardiomyocyte maturation by targeting Checkpoint kinase 1 (Chek1) [28,29]. The miR-15 family was significantly upregulated in 1D and 10D hearts based on microarray analysis. The miR-15 family is made up of six miRNAs (miR-15a, miR-15b, miR-16-1, miR-16-2, miR-195, and miR-497) that share significant homology and seed sequences. MiR-195 expression was the most upregulated between 1D and 10D, although miR-15a, miR-16, and miR-497 expression was also upregulated [28]. Two of three transgenic prenatal cardiomyocyte-specific (MHC) overexpression models resulted in neonatal cardiomyopathy and death. The single viable transgenic line had a significant reduction in heart weight and died at the age of 5–6 months due to slow-onset cardiomyopathy [28]. MiR-15-family loss-of-function after birth was associated with increased pH3+ cardiomyocytes and disorganized sarcomere structure but not with cardiomyocyte size. Furthermore, the authors observed Chek1 upregulation, confirming that the miR-15 family inhibits the expression of this kinase. These findings indicate that the miR-15 family was the first genuine miR implicated in the neonatal cardiomyocyte growth transition. 

Micro-RNAs’ roles in the heart are still being discovered and characterized. The miR-130 family, for example, includes miR-130a and miR-130b, which have been identified as cardiac development regulators due to repression of the transcriptional cofactor zinc-finger protein friend of GATA 2 (FOG-2). Furthermore, transgenic miR-130a overexpression results in ventricular hypoplasia and septal defects [166]. It was recently discovered that the expression levels of miR-29a, miR-30a, and miR-141 increase dramatically after birth. Antagomir treatment for these miRNAs increased the number of cycling cardiomyocytes and Cyclin A2 expression [167].

Most recently, it was shown that miR-294, which is expressed in the heart during embryonic development but lost in the adult heart, promotes cell cycle re-entry in both neonatal and adult cardiomyocytes [168]. Transiently introducing miR-294 into the heart after a myocardial infarction significantly improved cardiac structure and function. The beneficial effects of miR-294 have been linked to improved cell cycle re-entry, survival, angiogenesis, infarct size restriction, and induction of developmental signaling in the heart [168]. These findings point to miR-294 as a novel strategy for inducing pro-reparative changes in the heart.

Based on these collective findings of microRNA expression in the perinatal heart and their impact on cardiomyocyte proliferation, a model is proposed where microRNAs (in part) potentiate the transformation of proliferative fetal cardiomyocytes into hypertrophic postnatal cardiomyocytes (Figure 2) by transitioning cardiomyocytes from a proliferative growth (Hippo pathway) to physiological hypertrophic growth (IP3K/Akt/mTOR pathway) (Figure 2).

## 7. Conclusions and Future Directions

Understanding the molecular mechanisms that regulate the transition from a proliferative to a hypertrophic state in cardiomyocytes has significant therapeutic implications. By manipulating these pathways, it may be possible to stimulate cardiomyocyte proliferation and repair damaged heart tissue following myocardial infarction or other forms of heart disease.

Several strategies are currently being explored to achieve this goal. For example, small molecules that inhibit the Hippo pathway or activate YAP/TAZ have been shown to stimulate cardiomyocyte proliferation in vitro and in vivo. Similarly, inhibitors of mTOR have been shown to promote cardiomyocyte proliferation and improve heart function in animal models of heart disease.

However, these strategies also come with potential risks. For example, overactivation of YAP/TAZ or inhibition of mTOR can lead to uncontrolled cell proliferation and tumorigenesis. Therefore, it is crucial to develop strategies that can selectively stimulate cardiomyocyte proliferation without promoting tumorigenesis. One potential approach is to use targeted delivery systems that can deliver drugs specifically to cardiomyocytes. For example, nanoparticles or exosomes could be used to deliver small molecules or siRNAs that can modulate the activity of the Hippo or mTOR pathways, specifically in cardiomyocytes.

Another approach is to use gene therapy to modulate the expression of key regulators of cardiomyocyte proliferation. For example, adeno-associated viruses could be used to overexpress genes that promote cardiomyocyte proliferation or delete genes that inhibit proliferation. This approach has the advantage of being potentially more specific and long-lasting than small molecule-based therapies.

In conclusion, understanding the molecular mechanisms that regulate the transition from a proliferative to a hypertrophic state in cardiomyocytes is crucial for developing new therapies for heart disease. While significant progress has been made in this area, much work remains to be done. Future research should focus on elucidating the complex interplay between different signaling pathways in regulating cardiomyocyte proliferation and hypertrophy, as well as on developing safe and effective strategies to manipulate these pathways for therapeutic purposes.

## Figures and Tables

**Figure 1 cells-12-02324-f001:**
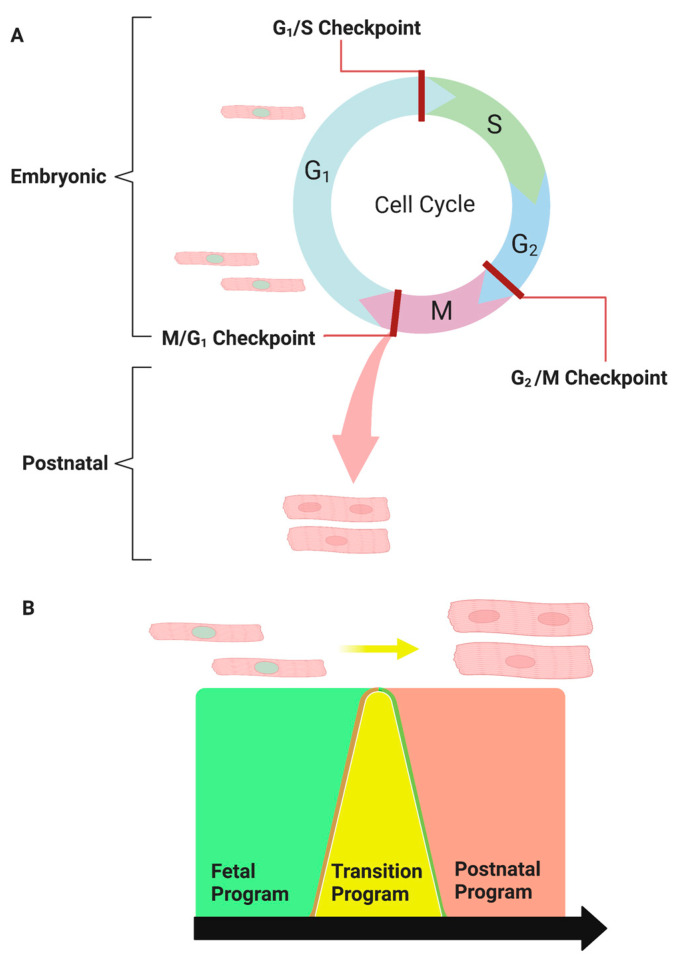
The transitional program: Reprogramming embryonic cardiomyocytes into postnatal cardiomyocytes. (**A**) Embryonic heart growth is primarily due to the high proliferative capacity of fetal cardiomyocytes, whereas postnatal heart growth is associated with binucleated cardiomyocyte hypertrophy. (**B**) Linking the fetal cardiogenomic program (green) to the postnatal cardiogenomic program (red). The transitional program (yellow) mediates the passage of proliferative fetal cardiomyocytes to exit the cell cycle prematurely to form binucleated hypertrophic cardiomyocytes.

**Figure 2 cells-12-02324-f002:**
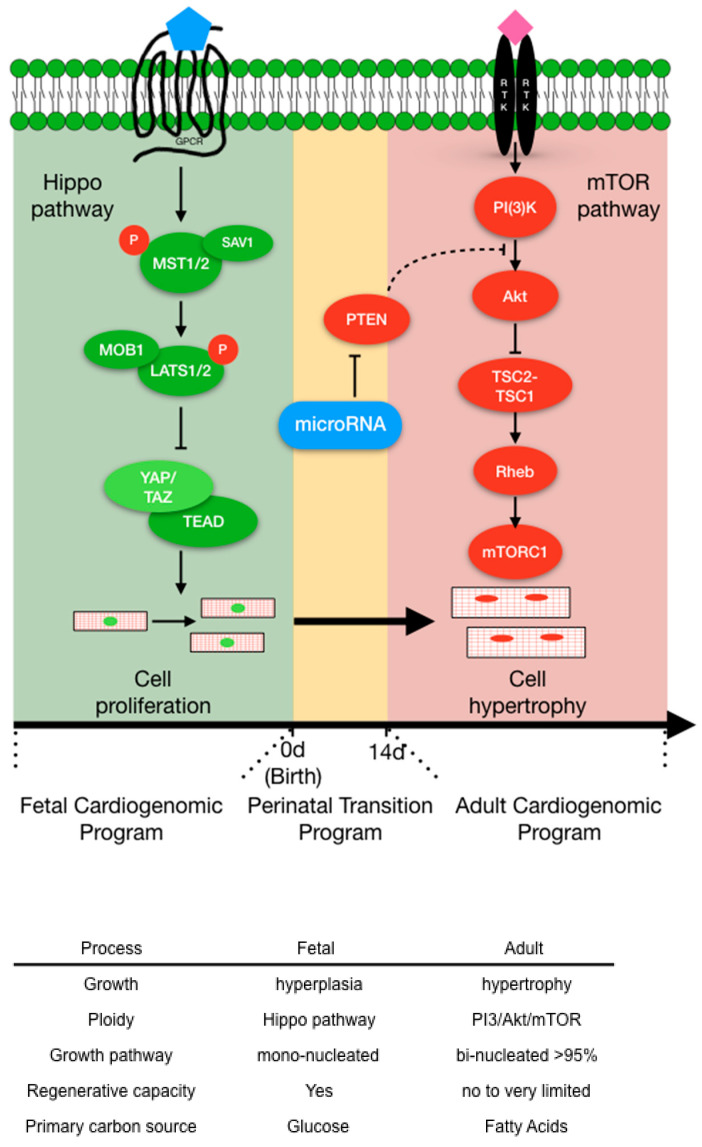
The transition of mammalian cardiomyocyte (CM) growth signaling pathways that convert proliferative fetal cardiomyocytes into hypertrophic adult cardiomyocytes. Top panel. The Hippo pathway is the primary fetal cardiomyogenic hyperplasia program (green) that regulates fetal CM number prior to birth (0 d). Birth triggers the perinatal transition program (yellow) that facilitates the conversion of fetal CM into adult hypertrophic CMs (red). MicroRNAs are excellent candidate molecules that propagate the CM perinatal transition. The mTOR pathway is the principal pathway that controls CM physiological hypertrophy. 0 d = time of birth and 14 d = 14 days post-birth. Bottom Panel: Highlights the differences between fetal (green) and adult (red) cardiomyocytes.

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
