# Peer review of "Regulatory Mechanisms That Guide the Fetal to Postnatal Transition of Cardiomyocytes"

_cells, 2023, doi:10.3390/cells12182324_

Round 1
Reviewer 1 Report
-This comprehensive review aims to summarize the recent advancements in understanding the transition program from proliferative to hypertrophic post-mitotic cardiomyocytes. The primary objective is to identify potential therapeutic targets that could facilitate effective myocardial repair and the restoration of structure and contractile function. The review is thoughtfully written, presenting a clear and detailed exploration of the mechanisms underlying cardiomyocyte proliferation. Its utility extends to cardiologists and researchers in the field, thus making it a suitable candidate for publication, following some necessary revisions. Below, I present my questions and concerns for the authors' consideration.
-While it is acknowledged that mechanisms such as adult cardiomyocyte proliferation/reprogramming and the involvement of resident/recruited stem cells (integral to myocardial regeneration (in regenerative medicine) represent the best path to restitutio ad integrum after cell death (e.g., infarction, acute myocarditis, acute toxicity, etc.), it must be recognized that these processes are often characterized by inefficiency and limited impact on the restoration of myocardial function. It is conceivable that the challenges lie in the complexity of promoting proliferation in post-mitotic tissue, as well as the complex endeavour of restoring tissue architecture (as evidenced by the failure of repair by using exogenous stem cells). Additionally, the intricate nature of retracing the morphogenetic program of myocardial histogenesis contributes to the impediments. Consequently, the observation of "restitutio ad integrum" by pathologists in necrotic myocardial regions is a rarity, with the formation of typical scar tissue is the prevailing outcome. A striking contrast can be drawn with instances where the activation of morphogenetic programs, as seen in zebrafish, salamanders, and other organisms, leads to a successful realization of restitutio ad integrum.
-Conversely, the repair of sublethal myocardial damage (e.g., chronic myocarditis, toxicity, endocrine deficiencies, aging, heart failure, etc.) occurs more frequently, exhibits enhanced efficiency (resulting in a rapid recovery of ejection fraction), and can be effectively managed through pharmacological interventions. Notably, many signaling pathways implicated in the transition program (such as mTOR, AKT, miR-133, various transcription factors, and sarcomeric gene targets) are similarly involved in the repair of sublethal damage. Unfortunately, this review does not delve into this alternative reparative approach, which has also garnered exploration.
-Notably absent from consideration are survival factors and the potential contributions of epicardial cells.
In conclusion, the current review exhibits promise and can be accepted after minor revisions.

Author Response
We want to thank the reviewer for their constructive feedback. Based on your suggestions, we have updated our introduction to include previous and current hypotheses of mammalian heart healing. In the title and abstract, we also toned down the therapeutic part of the review, as our primary goal is to emphasize the absence of extensive examination of the mammalian perinatal heart transition into the post-mitotic adult heart.
We contemplated including survival factors and epicardial cells in our present publication but believed they deserved to be reviewed separately.
Reviewer 2 Report
1# The title is “Transitioning from Fetal to Postnatal Cardiomyocyte States: An Update on Molecular Mechanisms and Therapeutic Approaches”. The author is requested to clearly address previous understandings and newly discovered knowledge regarding the molecular mechanism of myocyte transition.
Furthermore, the title is miss leading. The author did not explain any therapeutic approaches. They are suggested to create separate heading for the available therapeutic approaches. Finally, what would be the future possibilities of this approaches? It would be clearly depicted in this review.
2# The “Introduction” section is almost similar to the “Abstract” and very poor in size.
3# The Figure legend of Figure 2 is not satisfactory. It requires an elaborate description of the proposed drawing.
4# The main drawback of this manuscript is that it is completely narrative. The each signaling pathways demand a figure with appropriate labeling. The therapeutic approaches demand a Table with full of information.
5# The author cited 158 articles in this review manuscript. However, among the 158 only 32 articles are within previous 8 years. This represents the lack of updated information or low merit window of applied research. If possible, please upgrade the statistics of cited articles.
Author Response
We would like to thank the reviewer for their constructive insights. Based on your recommendations, we have revised our manuscript. We have responded to each of your suggestions below.
1# The title is “Transitioning from Fetal to Postnatal Cardiomyocyte States: An Update on Molecular Mechanisms and Therapeutic Approaches”. The author is requested to clearly address previous understandings and newly discovered knowledge regarding the molecular mechanism of myocyte transition.
Furthermore, the title is miss leading. The author did not explain any therapeutic approaches. They are suggested to create separate heading for the available therapeutic approaches. Finally, what would be the future possibilities of this approaches? It would be clearly depicted in this review.
Response: We modified the title to de-emphasize Therapeutic Approaches.
2# The “Introduction” section is almost similar to the “Abstract” and very poor in size.
Response: We have re-written the abstract and expanded the introduction to provide more context as to the rationale and primary focus of the review.
3# The Figure legend of Figure 2 is not satisfactory. It requires an elaborate description of the proposed drawing.
Response: Figure 2 legend is now more descriptive of the figure.
4# The main drawback of this manuscript is that it is completely narrative. The each signaling pathways demand a figure with appropriate labeling. The therapeutic approaches demand a Table with full of information.
Response: We believe the revised Figure 2 now provides sufficient information on the two key growth pathways, i) Hippo and ii) mTOR.
5# The author cited 158 articles in this review manuscript. However, among the 158 only 32 articles are within previous 8 years. This represents the lack of updated information or low merit window of applied research. If possible, please upgrade the statistics of cited articles.
Response: We have searched the literature for additional articles to include. We have added 10 additional references, with 9 of those citations from publications published in the past 8 years.
Round 2
Reviewer 2 Report
Removing Therapeutic Approaches from the manuscript is very easy attempt. However, providing a Table of therapeutic approaches defenitly increase the impact of this manuscript.
Author Response
We have addressed all comments of both reviewers in round 1. Adding a table of therapeutic approaches, as suggested by Reviewer 2, would obligate us to introduce text that expands on therapeutic techniques. While we agree that adding a therapeutic aspect might enhance interest in our review, we believe it would be speculative and premature to add to the current review that is focused on the perinatal heart.